# OpenReview forum: "Reranker Helps, but Not Enough: Towards Strong Poisoning Attacks Against Retrieval-Augmented Generation"
_ICML.cc/2026/Conference — ICML 2026 regular_

### Official Review · Reviewer_QcXK · 2026-02-21

**Soundness:** 3
**Presentation:** 3
**Significance:** 2
**Originality:** 2
**Overall Recommendation:** 3
**Confidence:** 4

**Summary:**

This paper studies a poisoning attack against reranker-enhanced RAG systems. The proposed method combines structured prompt manipulation with character-level perturbations to increase the likelihood that poisoned documents are ranked highly. The experiments show high attack success rates and resilience to several existing defenses, including paraphrasing, perplexity filtering, RobustRAG, and TrustRAG.

**Compliance With Llm Reviewing Policy:**

Affirmed.

**Final Justification:**

I will keep my original score unchanged.

**Key Questions For Authors:**

Please refer to the Weaknesses section for my detailed questions and concerns, particularly regarding model scope, real-world applicability, and defense evaluation.

**Limitations:**

No. Limitations are not discussed; if the authors cannot fully address the concerns raised in the weaknesses, they should explicitly acknowledge them in the limitations section.

**Strengths And Weaknesses:**

Strengths:

The experimental evaluation is extensive. The authors test across multiple datasets, retrievers, rerankers, and LLM generators, which gives a fairly complete picture of the attack’s behavior. The threat model is clearly articulated in both black-box and white-box settings, and the results consistently support the central claim: while rerankers mitigate earlier poisoning strategies, the proposed method is able to recover strong attack performance. Overall, the empirical section is careful and well executed.

Weaknesses:

The main limitation is the level of technical novelty. The rule-based prompt construction phase largely resembles prompt engineering designed to align with reranker scoring preferences. The character-level perturbation component builds on well-established gradient-based adversarial text attack techniques, and the use of beam search to maintain fluency is also standard. The overall pipeline is thoughtfully engineered and practically effective, but the technical contribution is incremental, as it is primarily in adapting and combining existing techniques for reranker-enhanced RAG, rather than introducing fundamentally new optimization or attack principles.

The evaluation scope is somewhat narrow with respect to models. Only two relatively dated rerankers (MiniLM and ELECTRA) are considered. In practice, many systems now rely on stronger cross-encoders or LLM-based rerankers. The transferability analysis is limited to the same two rerankers, which makes it difficult to assess how broadly the attack generalizes. Also, including stronger or closed-source models (e.g., ChatGPT or Gemini) would strengthen the claims about robustness and real-world relevance.

The real-world impact is also not fully explored. The experiments focus on benchmark-style QA with clearly specified questions and answers, whereas deployed systems often handle open-ended search queries, distribution shifts, and multi-turn interactions. It is unclear how sensitive the attack is to more natural user behavior. In addition, the paper does not analyze how feasible it would be for an attacker to insert poisoned documents into realistic corpora, nor does it explore the broader downstream consequences beyond incorrect answer generation.

Finally, the discussion of defenses is one-sided. The paper shows that several existing defenses are insufficient, but it doesn't explore mitigation strategies. Potential defenses such as adversarial training for rerankers, regularization at the ranking stage, or detection mechanisms are not investigated. A more balanced analysis of attack and defense would make the contribution stronger.

---

> ### Author Rebuttal · Authors · 2026-03-30
>
> > **Q1**: Evaluation scope: rerankers, transferability, and closed-source LLMs.
>
> We thank the reviewer for the suggestion and extend our evaluation along three aspects.
>
> 1. We include an additional experiment with BGE[1-2], a widely used and stronger reranker (0.6B parameters). As shown in Table 1, our method remains highly effective. This confirms that **our attack generalizes to stronger rerankers beyond MiniLM and ELECTRA**.
>
> ||Acc|ASR|
> |-|:-:|:-:|
> |Prompt Injection|48.8|5.8|
> |Disinformation|35.0|39.0|
> |PoisonedRAG-B|36.2|37.0|
> |PoisonedRAG-W|36.4|31.2|
> |CorruptRAG-AS|48.4|6.4|
> |CorruptRAG-AK|33.4|32.8|
> |P2A|12.0|78.4|
> |P3A|**8.6**|**83.0**|
>
> **Table 1. Performance on the NQ dataset using a BGE reranker and Llama3-8B.**
>
> 2. We further evaluate cross-reranker transfer by training on MiniLM or ELECTRA and testing on BGE. As shown in Table 2, the results suggest that **the attack does not overfit to a specific model**.
>
> ||Acc|ASR|
> |-|:-:|:-:|
> |MiniLM -> BGE|21.6|67.4|
> |ELECTRA -> BGE|19.0|68.4|
>
> **Table 2. Transfer performance of P3A on the NQ dataset. The model is trained on different rerankers and evaluated on BGE.**
>
> 3. We also evaluate stronger and closed-source LLMs as generators. As shown in Table 3, both P2A and P3A remain highly effective. This indicates that **stronger generation models do not mitigate the attack, highlighting its real-world relevance**.
>
> ||Llama3-8B||GPT-4o||Gemini2.5-Flash||
> |-|:-:|:-:|:-:|:-:|:-:|:-:|
> ||Acc|ASR|Acc|ASR|Acc|ASR|
> |P2A|15.8|63.0|18.4|58.6|14.2|68.4|
> |P3A|10.4|72.2|10.8|76.4|8.0|85.2|
>
> **Table 3. Performance on the NQ dataset with closed-source LLM generators.**
>
> [1] BGE M3-Embedding: Multi-Lingual, Multi-Functionality, Multi-Granularity Text Embeddings Through Self-Knowledge Distillation. ACL (2024).
>
> [2] https://huggingface.co/BAAI/bge-reranker-v2-m3.
>
> ---
> > **Q2**: The real-world impact is also not fully explored.
>
> We agree that open-ended queries and more realistic interaction settings are critical for assessing real-world impact.
>
> **Our current evaluation follows prior works[3-4] and focuses on the QA benchmark to ensure controlled and comparable measurement.
> Moreover, our method can be naturally extended to open-ended scenarios.**
> For a given open query, an attacker can first construct a biased answer or unsafe instruction as the target output.
> In our framework, the attacker then generates poisoned documents that present this target content in a credible and authoritative manner, ensuring both retrievability and influence on generation.
>
> We acknowledge that open-ended scenarios are not fully explored in this work. We consider these as important directions and will focus on them in future research.
>
> [3] PoisonedRAG: Knowledge Corruption Attacks to Retrieval-Augmented Generation of Large Language Models. USENIX Security (2025).
>
> [4] Machine Against the RAG: Jamming Retrieval-Augmented Generation with Blocker Documents. USENIX Security (2025).
>
> ---
> > **Q3**: Defense analysis: adversarial Training, regularization, and detection.
>
> Thanks for the suggestion, and we extend our study to include several mitigation strategies.
>
> 1. Following prior work[5-6], we use a reranker (Qwen3-Adv) trained by a mixture of adversarial methods, including HotFlip, PGD, and prompt injection. As shown in Table 4, P2A and P3A still achieve high ASR, indicating that **adversarial training provides limited robustness against our attacks**.
>
> ||Qwen3||Qwen3-Adv||
> |-|:-:|:-:|:-:|:-:|
> ||Acc|ASR|Acc|ASR|
> |P2A|3.2|91.4|3.0|90.8|
> |P3A|5.4|86.4|5.8|86.6|
>
> **Table 4. Performance on the NQ dataset using a standard reranker (Qwen3) and its adversarially trained variant (Qwen3-Adv).**
>
> 2. We explore a ranking-time regularization based on Maximal Marginal Relevance (MMR)[7-8]. The intuition is that poisoned documents often share high similarity, and MMR penalizes redundancy while preserving relevance, which reduces the likelihood that multiple similar poisoned documents are selected. As shown in Table 5, **MMR yields only a marginal reduction in ASR**.
>
> ||w/o Reg||Reg||
> |-|:-:|:-:|:-:|:-:|
> ||Acc|ASR|Acc|ASR|
> |NQ|10.4|72.2|17.4|71.8|
> |MS-MARCO|15.6|53.0|19.0|49.8|
> |HotpotQA|13.0|72.8|20.4|68.0|
>
> **Table 5. Performance of P3A with and without MMR regularization.**
>
> 3. We have reported perplexity-based detection in Figure 3 of the paper, and further extended it to stronger LLM-based detectors (**see our response to Reviewer `RGe7` Q2**).
>
> [5] Unifying Adversarial Robustness and Training Across Text Scoring Models.
>
> [6] https://huggingface.co/Qwen/Qwen3-Reranker-0.6B.
>
> [7] The use of MMR, diversity-based reranking for reordering documents and producing summaries. SIGIR (1998).
>
> [8] Joint passage ranking for diverse multi-answer retrieval. EMNLP (2021).
>
> ---
> Thank you again for the helpful feedback and thoughtful guidance. We hope our detailed responses successfully resolve your concerns. If there are any remaining uncertainties, we would be happy to clarify them.

---

> > ### Author Rebuttal · Reviewer_QcXK · 2026-04-04
> >
> > Thank you for the rebuttal and additional experiments. While the expanded model and defense evaluation improves the paper’s completeness, I still feel the real-world impact and deployment implications are not fully addressed and would benefit from deeper discussion, so I will keep my original score unchanged.

---

> > > ### Author Response · Authors · 2026-04-04
> > >
> > > We are glad that our rebuttal has resolved your concerns on expanded models and defense analysis. We would like to further clarify the remaining point regarding real-world impact.
> > >
> > > While we agree that open-ended scenarios are interesting, **we respectfully argue that focusing on QA benchmarks is the standard evaluation protocol used in prior RAG poisoning work [1-5]**. This ensures controlled and comparable measurement. Therefore, QA evaluation is a scope choice shared by existing work, rather than a limitation unique to our paper.
> > >
> > > Furthermore, as detailed in our previous response, **our method can naturally extend to open-ended scenarios**. An attacker can specify a biased answer or unsafe instruction as the target, and then construct poisoned documents that are both retrievable and persuasive for generation.
> > >
> > > We hope the reviewer can assess the paper based on its central contribution: **(1) identifying a realistic and previously underexplored vulnerability in reranker-enhanced RAG, and (2) supporting this claim with substantially expanded evaluation and defense analysis**.
> > >
> > > If you find this clarification sufficient, we would be very grateful if you could reconsider your score. Your constructive feedback has improved our paper, and your support means a great deal to us.
> > >
> > > ---
> > >
> > > [1] PoisonedRAG: Knowledge Corruption Attacks to Retrieval-Augmented Generation of Large Language Models. USENIX Security (2025).
> > >
> > > [2] Machine Against the RAG: Jamming Retrieval-Augmented Generation with Blocker Documents. USENIX Security (2025).
> > >
> > > [3] Practical Poisoning Attacks against Retrieval-Augmented Generation. ACM SACMAT (2026).
> > >
> > > [4] Traceback of Poisoning Attacks to Retrieval-Augmented Generation. ACM WWW (2025).
> > >
> > > [5] TrustRAG: Enhancing Robustness and Trustworthiness in Retrieval-Augmented Generation.

---

### Official Review · Reviewer_RGe7 · 2026-03-02

**Soundness:** 3
**Presentation:** 3
**Significance:** 3
**Originality:** 3
**Overall Recommendation:** 5
**Confidence:** 4

**Summary:**

Authors introduce a novel framework, P^3A (Prompt-Perturbation Poisoning Attack), in order to evaluate the vulnerability of reranker-enhanced RAG systems to data poisoning. Using GPT-2, authors compute perplexity scores to demonstrate that fine-tuning a reranker on benign in-domain data serves as an effective simple and cheap defense, significantly degrading the performance of existing targeted poisoning attacks. P^3A works as a two-phase framework: where the initial phase employs rule-based prompt engineering to generate adversarial documents that are both semantically coherent and aligned with reranker preferences; in the second phase, authors apply character-level perturbations, using optimization inspired by projected gradient descent (PGD) and beam search to maximize reranker scores while introducing minimal textual changes. Athors also performed a benchmark of attack methods on three datasets (NQ, MS-MARCO, HotpotQA) across different rerankers (MiniLM, ELECTRA) and LLMs;  demonstrating that their presented framework substantially outperforms prior attack methods.

**Compliance With Llm Reviewing Policy:**

Affirmed.

**Final Justification:**

Thank you for the rebuttal. My concerns are resolved. In particular, the additional evaluation with stronger language-model-based detectors is helpful and directly addresses my question about whether the GPT-2-based detection result generalizes. I am changing my score

**Key Questions For Authors:**

Have the authors considered evaluating against more capable language-model-based detectors ?

**Limitations:**

Author are partially addressed, authors state that while both defenses reduce ASR for prior attacks to some extent, their impact on P^2A
and P^3A remains limited.

**Strengths And Weaknesses:**

The paper's core empirical finding (fine-tuning a reranker on benign data provides a robust defense against existing poisoning attacks) is addequately supported by the experimental results presented in Table 1 and Figure 1. The two-phase attack framework is methodologically well-founded: the rule-based phase builds upon prior analysis of reranker preferences, while the perturbation pipeline forms a coherent and integrated optimization strategy.  One issue I found is that  there is no ablation studies or direct analysis of the mechanism used by rerankers to filter poisoned documents. Also, the evaluation of perplexity-based detection relies exclusively on GPT-2, leaving it unclear whether more capable language-model-based detectors would prove more effective.

---

> ### Author Rebuttal · Authors · 2026-03-30
>
> > **Q1**: There is no ablation studies or direct analysis of the mechanism used by rerankers to filter poisoned documents.
>
> Thanks for the suggestion. We provide an additional analysis to quantify how rerankers affect poisoned documents.
>
> As shown in Table 1, we measure the number of adversarial documents appearing in the final Top-K results. Without a reranker, most poisoned documents are retained. **After applying the reranker, this number drops substantially, indicating that rerankers push poisoned documents to lower ranks and thereby reduce attack effectiveness.** In contrast, P2A and especially P3A maintain a higher presence in Top-K, explaining their stronger performance.
>
> **We would like to clarify that the current manuscript includes a discussion of the underlying mechanism in Section 4.** We attribute this behavior to the reranker’s cross-attention architecture and fine-grained semantic evaluation, which allow it to detect incoherent or artificially constructed passages.
>
> We agree that more detailed ablation studies and interpretability analyses would further strengthen the understanding. We leave such analyses for future work.
>
> |||w/o Reranker|Reranker|
> |-|-|:-:|:-:|
> |NQ|PoisonedRAG-B|4.57|1.18|
> ||CorruptRAG-AS|4.66|0.06|
> ||P2A|4.87|2.80|
> ||P3A|4.81|4.21|
> |MS-MARCO|PoisonedRAG-B|3.54|0.89|
> ||CorruptRAG-AS|4.55|0.01|
> ||P2A|4.36|1.62|
> ||P3A|4.15|2.82|
> |HotpotQA|PoisonedRAG-B|5.00|1.64|
> ||CorruptRAG-AS|5.00|0.17|
> ||P2A|5.00|4.09|
> ||P3A|5.00|4.44|
>
> **Table 1. Average number of adversarial documents in Top-K results (K=5) with and without a reranker (MiniLM). Each query is associated with 5 injected poisoned documents.**
>
> ---
> > **Q2**: Have the authors considered evaluating against more capable language-model-based detectors?
>
> We extend the detection to stronger language models, including Llama3-8B and Qwen2.5-7B.
>
> As shown in Table 2, malicious documents consistently exhibit higher mean perplexity than clean ones across all models. However, the increase is relatively small, and the distributions still exhibit substantial overlap.
>
> This is further reflected in detection metrics. Although some settings show moderate AUROC improvements, the performance remains far below the level required for reliable deployment. In practice, AUROC ≥ 0.95 is typically needed for robust detection, which is not achieved in any configuration.
>
> Overall, these findings suggest that **stronger language models alone are insufficient for effective detection**, highlighting the need for more specialized approaches.
>
> ||GPT-2||||Llama3-8B||||Qwen2.5-7B||||
> |-|:-:|:-:|:-:|:-:|:-:|:-:|:-:|:-:|:-:|:-:|:-:|:-:|
> ||Mean-C|Mean-M|AUROC|AUPRC|Mean-C|Mean-M|AUROC|AUPRC|Mean-C|Mean-M|AUROC|AUPRC|
> |NQ|31.71|55.45|0.805|0.780|10.47|16.83|0.796|0.736|9.47|15.10|0.762|0.712|
> |MS-MARCO|44.89|65.48|0.730|0.698|14.71|25.96|0.773|0.755|12.96|27.73|0.822|0.809|
> |HotpotQA|30.79|39.96|0.660|0.513|8.24|16.60|0.859|0.731|7.80|15.96|0.832|0.717|
>
> **Table 2. Perplexity-based detection using different language models. We report the mean perplexity of clean (Mean-C) and malicious (Mean-M) documents, along with AUROC and AUPRC.**
>
> ---
> Thank you again for the helpful feedback and thoughtful guidance. We hope our detailed responses successfully resolve your concerns. If there are any remaining uncertainties, we would be happy to clarify them.

---

> > ### Author Rebuttal · Reviewer_RGe7 · 2026-04-03
> >
> > Thank you for the rebuttal. My concerns are resolved. In particular, the additional evaluation with stronger language-model-based detectors is helpful and directly addresses my question about whether the GPT-2-based detection result generalizes.

---

> > > ### Author Response · Authors · 2026-04-03
> > >
> > > We are very glad that our rebuttal has fully addressed your concerns. We sincerely appreciate your positive feedback. If you feel it is appropriate, we would be very grateful if you could consider raising your score. Your constructive feedback has improved our paper, and your support means a great deal to us.

---

### Official Review · Reviewer_ag3S · 2026-03-11

**Soundness:** 3
**Presentation:** 3
**Significance:** 2
**Originality:** 2
**Overall Recommendation:** 4
**Confidence:** 4

**Summary:**

This paper proposes poisoning attacks against RAG. First, they suggest $P^2A$, which is based on prompt design principles that the authors have identified. $P^3A$ is a character-level perturbation method with a white-box RAG, using candidate generation through a gradient descent approach. Then, beam search refinement is applied, and the final results are collected using a predefined reranker score $\tau$.

**Compliance With Llm Reviewing Policy:**

Affirmed.

**Final Justification:**

I would like to keep my final score as weak accept. I believe this work is a combination of prompt template engineering and character perturbations. I don't think this work has a high impact on the sub-area to warrant a score of 5, as it particularly targets only reranker RAG, which limits the scope of this work. The approach is reasonable and solid, based on the authors' responses during the rebuttal. So I will keep my score as weak accept.

**Key Questions For Authors:**

**Questions**

* Based on my understanding, this approach only holds for reranker-enhanced RAG, which is a fairly limited scope.
* Furthermore, based on Table 2, the rule-based approach already has an impressive ASR compared to baselines. If the authors apply character-level perturbation to CorruptRAG, does this also improve performance? What is the main contribution or finding of $P^3A$?
* Is the rule-based prompt phase dependent on the LLM used for rephrasing?

**Strengths And Weaknesses:**

**Strengths**
* It demonstrates impressive improvements in poisoning attacks against reranker-enhanced RAG and defenses.
* It is well described and technically sound.
* It contains various experiments to demonstrate the proposed approaches on different datasets and models.

**Weaknesses**
* It is not fully described how the authors identified the rules and how each rule affects the final poisoning attacks. Which tone is most important, or how strong an LLM is needed, is not fully analyzed.
* Since $P^3A$ employs a white-box reranker, it would be helpful to compare transfer attacks with baselines (it would be good to have both Tables 2 and 7 in the same table). Specifically, using MiniLM as a reranker and evaluating the poisoning attack success rate on different rerankers (e.g., Electra), and vice versa, across different models and datasets.
* One of the main weaknesses is that the current approach uses a white-box reranker and is specifically targeted at reranker-enhanced RAG. It does not show the same improvement in vanilla RAG, which suggests that the approach is limited to reranker-enhanced RAG applications.

---

> ### Author Rebuttal · Authors · 2026-03-30
>
> > **Q1**: This approach only holds for reranker-enhanced RAG, which is a fairly limited scope.
>
> We would like to clarify that our approach is not limited to reranker-enhanced RAG.
> We focus on reranker-enhanced RAG because rerankers are widely adopted in modern RAG systems and have a substantial impact on attack effectiveness, *yet prior work largely overlooks their role*.
>
> **In the original manuscript, the results on vanilla RAG have also been provided in Figure 1 and Table 3, where both P2A and P3A remain highly effective.**
> These results demonstrate that our method generalizes across different RAG pipelines.
>
> ---
> > **Q2**: Does adding character level perturbations further improve baselines, and what is the main contribution of P3A?
>
> Thanks for the suggestion, we conduct an additional experiment by applying our character-level perturbation phase to CorruptRAG-AS.
>
> 1. As shown in Table 1, perturbation significantly improves the attack strength. The ASR increases from 3.0% to 42.4%, indicating that **the perturbation phase is a flexible and plug-and-play technique that can enhance prior methods**.
>
> ||Acc $\downarrow$|ASR $\uparrow$|
> |-|:-:|:-:|
> |CorruptRAG-AS|36.6|3.0|
> |CorruptRAG-AS+|28.2|42.4|
>
> **Table 1. Effect of applying character-level perturbation to prior attacks. CorruptRAG-AS+ denotes CorruptRAG-AS enhanced with our perturbation phase, using the MiniLM reranker.**
>
> 2. Table 2 reveals a minor limitation: the optimized text in CorruptRAG-AS+ exhibits a much larger edit distance and change ratio compared to P3A. **This leads to noticeable degradation in readability and makes the attack more detectable**.
>
> These results highlight our main contribution. While character-level perturbation alone can improve existing attacks, it lacks control over semantic coherence. **P3A integrates a rule-based prompt phase with perturbation, ensuring both high attack effectiveness and minimal textual distortion**. We will add these results in the revision.
>
> ||Edit Distance|Change Ratio|
> |-|:-:|:-:|
> |P3A|5.33|1.32%|
> |CorruptRAG-AS+|19.98|4.98%|
>
> **Table 2. Average Levenshtein distance and change ratio relative to the poisoned texts before and after the character-level perturbation phase.**
>
> ---
> > **Q3**: Is the rule-based prompt phase dependent on the LLM used for rephrasing?
>
> We evaluate the rule-based prompt phase using different LLMs for generating poisoned texts, including GPT-4o and Gemini2.5-Flash.
>
> As shown in Table 3, both LLMs achieve strong ASR on all datasets.
> These results indicate that **the rule-based prompt phase does not depend on a specific LLM**. The effectiveness mainly comes from the structured prompting strategy rather than model-specific behavior.
>
> ||GPT-4o||Gemini2.5-Flash||
> |-|:-:|:-:|:-:|:-:|
> ||Acc $\downarrow$|ASR $\uparrow$|Acc $\downarrow$|ASR $\uparrow$|
> |NQ|15.8|63.0|16.6|57.4|
> |MS-MARCO|21.2|40.6|23.0|36.6|
> |HotpotQA|13.4|73.2|16.8|63.4|
>
> **Table 3. Performance of P2A across datasets using different LLMs in the rule-based prompt phase.**
>
> ---
> Thank you again for the helpful feedback and thoughtful guidance. We hope our detailed responses successfully resolve your concerns. If there are any remaining uncertainties, we would be happy to clarify them.

---

> > ### Author Rebuttal · Reviewer_ag3S · 2026-04-06
> >
> > I assume authors missed the raised weakness and only respond to my questions.
> > * It is not fully described how the authors identified the rules and how each rule affects the final poisoning attacks. Which tone / prompt is most important, or how strong an LLM is needed, is not fully analyzed.
> > * Since  employs a white-box reranker, it would be helpful to compare transfer attacks with baselines (it would be good to have both Tables 2 and 7 in the same table). Specifically, using MiniLM as a reranker and evaluating the poisoning attack success rate on different rerankers (e.g., Electra), and vice versa, across different models and datasets.
> > * One of the main weaknesses is that the current approach uses a white-box reranker and is specifically targeted at reranker-enhanced RAG. It does not show the same improvement in vanilla RAG, which suggests that the approach is limited to reranker-enhanced RAG applications.

---

> > > ### Author Response · Authors · 2026-04-07
> > >
> > > We feel deeply sorry for the oversight in the previous response. We mistakenly treat the "Key Questions" as the main concerns underlying the weaknesses, as they appear highly correlated (e.g., W1 aligns with Q3, W3 with Q1). Thanks for your kind reminder, we provide further clarifications below.
> > >
> > > > **W1**: How the authors identified the rules, how each rule affects the final poisoning attacks, and how strong an LLM is needed.
> > >
> > > 1. The prompt rules are derived from reranker preference bias. **Since rerankers are trained on encyclopedic data (e.g., Wikipedia) [1-2], they favor authoritative, semantically dense texts.** By exploiting this, we formulate linguistic principles that align poisoned texts with these distributional preferences, maximizing relevance scores.
> > >
> > > 2. To evaluate the contribution of each prompt rule, we conduct an ablation study. **As shown in Table 1, all rules contribute to the overall attack effectiveness.** Notably, removing "Rule 1: Directly State the Answer" causes the most significant drop in ASR, indicating that **explicitly presenting the target answer is the most important factor for steering the generator**. "Rule 4: Maintain Sharp Focus" also exhibits a substantial impact, underscoring its importance in preserving high semantic density. We will add these results in the revision.
> > >
> > > ||Acc $\downarrow$|ASR $\uparrow$|
> > > |-|:-:|:-:|
> > > |P2A|15.8|63.0|
> > > |w/o #1. Directly State the Answer|19.2|54.8|
> > > |w/o #2. Use an Authoritative Tone|16.6|61.2|
> > > |w/o #3. Provide Supporting Context|16.6|60.8|
> > > |w/o #4. Maintain Sharp Focus|17.4|57.8|
> > >
> > > **Table 1. Ablation study of prompt rules for P2A on the NQ dataset, using MiniLM and Llama3-8B.**
> > >
> > > 3. In our previous response to Q3, additional experiments show that **P2A remains effective when the poisoned texts are generated by different LLMs**. This confirms that **our method does not require a highly advanced LLM, and that the attack effectiveness is primarily driven by the prompting strategy itself**. Please refer to Q3 for the complete discussion.
> > >
> > > [1] MiniLM: Deep Self-Attention Distillation for Task-Agnostic Compression of Pre-Trained Transformers. NeurIPS (2020).
> > >
> > > [2] ELECTRA: Pre-training Text Encoders as Discriminators Rather Than Generators. ICLR (2020).
> > >
> > > > **W2**: It would be helpful to compare transfer attacks with baselines.
> > >
> > > We agree it would be helpful to compare our transfer attack (P3A-T) against baselines. As shown in Table 2, **P3A-T outperforms all baselines across different rerankers**. By optimizing perturbations on a surrogate model (BGE [3-4]), the poisoned texts maintain strong adversarial effectiveness, achieving high ASR (66.4% on MiniLM and 68.0% on ELECTRA). These results confirm that **transfer attacks effectively bypass unknown rerankers, highlighting the practical threat P3A-T poses to real-world RAG deployments**. We will add these results in the revision.
> > >
> > > ||MiniLM|||ELECTRA|||
> > > |-|:-:|:-:|:-:|:-:|:-:|:-:|
> > > ||Acc $\downarrow$|ASR $\uparrow$|F1-Score $\uparrow$|Acc $\downarrow$|ASR $\uparrow$|F1-Score $\uparrow$|
> > > |Prompt Injection|37.4|2.4|0.0008|41.2|4.6|0.0020|
> > > |Disinformation|30.2|27.6|0.1820|33.8|30.2|0.2068|
> > > |PoisonedRAG-B|28.6|34.8|0.2368|34.0|31.6|0.2328|
> > > |PoisonedRAG-W|31.8|15.8|0.1048|30.4|21.0|0.1592|
> > > |CorruptRAG-AS|36.6|3.0|0.0128|39.2|7.0|0.0356|
> > > |CorruptRAG-AK|28.2|27.2|0.1656|33.4|27.4|0.1748|
> > > |P2A|15.8|63.0|0.5592|17.8|66.8|0.5988|
> > > |P3A|10.4|72.2|0.8416|14.2|71.0|0.7796|
> > > |P3A-T|14.0|66.4|0.7746|16.6|68.0|0.6462|
> > >
> > > **Table 2. Performance on the NQ dataset using Llama3-8B. P3A-T denotes poisoned texts optimized on the BGE reranker and evaluated on different target rerankers.**
> > >
> > > [3] BGE M3-Embedding: Multi-Lingual, Multi-Functionality, Multi-Granularity Text Embeddings Through Self-Knowledge Distillation. ACL (2024).
> > >
> > > [4] https://huggingface.co/BAAI/bge-reranker-v2-m3.
> > >
> > > > **W3**: The current approach uses a white-box reranker and is specifically targeted at reranker-enhanced RAG. It does not show the same improvement in vanilla RAG.
> > >
> > > We agree that P3A does not exhibit the same improvement in vanilla RAG. This occurs because the perturbations are optimized to exploit the semantic scoring of rerankers. **Without a reranker, these perturbations bring little additional benefit and may slightly weaken retrieval quality.** However, **both P2A and P3A still achieve high ASR in the vanilla setting**, remaining highly competitive with baselines.
> > >
> > > Deploying a reranker has become standard in production-grade RAG systems to ensure high retrieval precision. **As existing poisoning attacks are mitigated by this component, targeting reranker-enhanced RAG reveals a critical security vulnerability.** Therefore, our approach is highly relevant, exposing risks in widely adopted RAG architectures.
> > >
> > > ---
> > >
> > > We hope our follow-up response could address your remaining concerns. We sincerely appreciate your constructive feedback, and your support means a great deal to us. If you have any further questions, please feel free to let us know.

---

### Official Review · Reviewer_WftZ · 2026-03-13

**Soundness:** 3
**Presentation:** 3
**Significance:** 3
**Originality:** 3
**Overall Recommendation:** 5
**Confidence:** 3

**Summary:**

The paper introduces P3A to poison RAG systems that utilize a reranker. P3A operates in two stages: Rule-based Prompt and Character-level Perturbation. It injects subtle optimized character-level perturbations into poison data. It can achieve high attack success rates while remaining natural and readable.

**Compliance With Llm Reviewing Policy:**

Affirmed.

**Final Justification:**

The authors have effectively demonstrated that the attack is robust against common text-smoothing defenses and remains effective even when lexical matching (BM25) is employed. The clarification on black-box transferability successfully mitigates my concerns regarding the white-box threat model.

**Key Questions For Authors:**

Have you tried it on sparse retriever like BM25? Since these do not use the same semantic cross-attention as neural rerankers, how would the attack's effectiveness change?

Could automated smoothing tools (e.g., Grammarly) or simple spell-checkers neutralize the character-level perturbations by correcting the "optimized" typos?

**Limitations:**

Yes

**Strengths And Weaknesses:**

Strengths:

\+ The low-level gradient-based perturbations is a novel approach. The combination of high-level linguistic strategies with low-level character perturbations is a sophisticated method for bypassing rerankers. By targeting the "blind spots" in semantic cross-attention mechanisms, the authors expose vulnerabilities that traditional retrieval-only attacks miss.

\+ The evaluation uses g three diverse datasets and multiple model architectures against several strong baselines.

\+ P3A outperforms strong baselines like PoisonedRAG and CorruptRAG.

Weaknesses:

-- The attack relies on a white-box access to the reranker's parameters and gradients, which may not always be available to an external adversary.

\- The linguistic pattern may work only for limited languages. The four principles for prompt engineering are derived from patterns in English-language rerankers.

\- The effectiveness of these patterns in other languages or against sparse retrievers like BM25, which rely on keyword matching rather than semantic attention, remains a question.

\- The authors test against a paraphrasing defense, the attack might be vulnerable to simpler smoothing methods, such as grammar checkers or spell-checkers.

---

> ### Author Rebuttal · Authors · 2026-03-30
>
> > **Q1**: The attack assumes white-box access to the reranker, which may not hold in practice.
>
> We agree that assuming white-box access to the reranker is less realistic, but we would like to clarify that our method does not rely solely on white-box access.
>
> 1. **P2A is a fully black-box method that requires no access to reranker parameters or gradients**. As shown in Table 2, P2A already surpasses all baselines and achieves strong attack performance, demonstrating that effective poisoning is feasible under realistic black-box settings.
> 2. We evaluate the transferability of P3A in Table 7. The results show that attacks optimized on one reranker remain effective on unseen rerankers, with consistently high ASR. This indicates that **even when direct access is unavailable, an attacker can leverage a surrogate model to perform the attack**.
>
> ---
> > **Q2**: The proposed prompt principles may be limited to English and not generalize across languages.
>
> We would like to clarify the four principles are not tied to any specific language, but are instead derived from general properties of reranker training data.
>
> In particular, rerankers are trained to favor passages that are relevant, coherent, and informative with respect to a query. The principles reflect these universal preferences, rather than language-specific patterns. Therefore, **our method can be naturally extended to other languages by generating poisoned texts that follow the same principles in the target language**.
>
> ---
> > **Q3**: How effective is the attack against sparse retrievers such as BM25?
>
> Thanks for the suggestion, we extend our evaluation to a sparse retriever (BM25) and report the results.
>
> Compared with the dense retriever (Contriever), the attack remains effective on BM25, although with a moderate reduction in ASR.
> This gap is expected, as BM25 relies on lexical matching rather than semantic cross-attention, making it less sensitive to the semantic coherence exploited by our attack.
>
> Overall, **the attack still achieves strong effectiveness, confirming its robustness across different retrieval paradigms**. We will add these results to the revision.
>
> |||Contriever||BM25||
> |-|-|:-:|:-:|:-:|:-:|
> |||Acc $\downarrow$|ASR$\uparrow$|Acc$\downarrow$|ASR$\uparrow$|
> |NQ|PoisonedRAG-B|28.6|34.8|33.6|7.8|
> ||P2A|15.8|63.0|20.6|40.8|
> ||P3A|10.4|72.2|17.8|48.6|
> |MS-MARCO|PoisonedRAG-B|28.8|24.2|29.8|3.6|
> ||P2A|21.2|40.6|23.0|35.0|
> ||P3A|15.6|53.0|18.6|44.4|
> |HotpotQA|PoisonedRAG-B|30.2|35.0|36.8|21.2|
> ||P2A|13.4|73.2|17.0|64.4|
> ||P3A|13.0|72.8|17.0|67.6|
>
> **Table 1. Performance on dense (Contriever) and sparse (BM25) retrievers using a MiniLM reranker and Llama3-8B.**
>
> ---
> > **Q4**: Could automated smoothing tools (e.g., Grammarly) neutralize the character-level perturbations by correcting the "optimized" typos?
>
> We evaluate whether automated correction tools can neutralize the character-level perturbations by applying GECToR[1-3].
>
> As shown in Table 2, the results show that **automatic spell-checking and text smoothing have a limited impact on attack effectiveness**.
>
> In the worst case, the performance degrades toward the P2A regime, which remains highly effective as shown in our paper.
>
> |||Acc $\downarrow$|ASR $\uparrow$|
> |-|-|:-:|:-:|
> |NQ|P3A|10.4|72.2|
> ||P3A-C|12.8|77.6|
> |MS-MARCO|P3A|15.6|53.0|
> ||P3A-C|20.8|46.2|
> |HotpotQA|P3A|13.0|72.8|
> ||P3A-C|13.0|72.2|
>
> **Table 2. Attack performance before and after grammatical error correction. P3A-C denotes the corrected version using GECToR.**
>
> [1] GECToR – Grammatical Error Correction: Tag, Not Rewrite. Proc. BEA (2020).
>
> [2] Text Simplification by Tagging. Proc. BEA (2021).
>
> [3] https://github.com/grammarly/gector.
>
> ---
> Thank you again for the helpful feedback and thoughtful guidance. We hope our detailed responses successfully resolve your concerns. If there are any remaining uncertainties, we would be happy to clarify them.

---

> > ### Author Rebuttal · Reviewer_WftZ · 2026-04-03
> >
> > Thanks for your response. My concerns are addressed. I will consider raising my score.

---

> > > ### Author Response · Authors · 2026-04-03
> > >
> > > We are very glad that our rebuttal has fully addressed your concerns. We sincerely appreciate your willingness to consider raising your score. Your constructive feedback has helped improve our paper, and your support means a great deal to us.

---

### Decision · Program_Chairs · 2026-04-30

**Decision:**

Accept (regular)

**Comment:**

The paper studies poisoning attacks against reranker-enhanced RAG systems and proposes a two-stage framework, P3A, that first uses rule-based prompt construction to produce poisoned documents aligned with reranker preferences, and then applies subtle character-level perturbations to further increase their ranking scores while preserving readability and adversarial effectiveness. The reviewers generally agreed that the paper addresses an important and practically relevant problem, namely the vulnerability of reranker-enhanced RAG systems to poisoning, and found the empirical evaluation extensive and carefully executed across multiple datasets, rerankers, retrievers, generators, and defenses.

There are some limitations of this paper. Some reviewers noted that the technical novelty is incremental, as the framework mainly combines prompt engineering with character-level adversarial perturbation techniques, and others pointed out that the model scope remains somewhat narrow, especially with evaluation focused on a small number of rerankers. There were also requests for more analysis of how the rules were derived, broader transfer evaluation, and more discussion of real-world applicability and defenses.

Overall, I find this to be a technically solid and well-evaluated paper with clear practical relevance. While the methodological novelty is somewhat incremental and the scope could be broadened further, the paper makes a useful contribution by identifying a meaningful blind spot in reranker-based defenses and by providing a strong empirical study of this failure mode. I therefore recommend acceptance.